# SIFTING THE SIGNAL FROM THE NOISE

**Daniel A. Herrmann & Jacob VanDrunen**

## ABSTRACT

Signaling games are useful for understanding how language emerges. In the standard models the dynamics in some sense already knows what the signals are, even if they do not yet have meaning. In this paper we develop a simple model we call an *attention game* in which agents have to learn which feature in their environment is the signal. We demonstrate that simple reinforcement learning agents can still learn to coordinate in contexts in which (i) the agents do not already know what the signal is and (ii) the other features in the agents' environment are uncorrelated with the signal. Furthermore, we show that, in the cases in which other features are correlated with the signal, there is a surprising trade-off between learning to pay attention to the signal and success in action. We show that the mutual information between a signal and a feature plays a key role in governing the accuracy and attention of the agent.

## 1 INTRODUCTION

Lewis-Skyrms signaling games are useful for understanding how language emerges (Lewis, 1969; Skyrms, 2010). Nature presents some stimulus which yields a payoff if a particular action is taken in response. The *sender* observes the stimulus, and sends some *signal* to the *receiver*. The receiver in turn observes the signal and performs some action. If the action corresponds to the state of nature, the agents receive the payoff. In Skyrms's paradigm, what is of principal interest is whether agents will converge to a strategy profile which maximizes information transfer when their dispositions are subject to an adaptive process such as reinforcement learning.

For the case of 2-sender, 2-signal, 2-act ($2 \times 2 \times 2$) signaling games under simple reinforcement,[1] Argiento et al. (2009) proved that convergence to an optimal signaling system is guaranteed in the limit. This means that, in this simple context, agents will always learn a signaling system. This learning procedure assumes that the receiver pays attention to the signal channel. But in the real world, the receiver might not know what part of the act that the sender performs is meant to be the signal. The collection of models we provide here addresses this problem: how might agents learn which available stimulus is best to condition their actions on. This is a fundamental concern for the theory of self-assembling games (Barrett & Skyrms 2017).

Our work draws inspiration from other models in the literature. Herrmann and Skyrms (forthcoming) provide a model of the invention of conventions in which agents need to learn the properties on which they condition their strategies. In the epistemic network game of Barrett et al. (2019), agents learn to attend to other agents. Barrett (2020) gives a Lewis-Skyrms signaling game model in which agents must learn to distinguish between a signal with an already-established meaning and an uncorrelated feature of the world when deciding how to condition their actions.[2]

## 2 ATTENTION GAMES

Our *attention games* extend the Lewis-Skyrms signaling game. In the traditional $2 \times 2 \times 2$ Lewis signaling game, the receiver observes only whether one signal $s$ or the other is sent. Since it is

---

[1] This is a special case of a learning process commonly used in psychology and economics (Luce 1959; Herrnstein 1970; Erev & Roth 1998). For an analysis of its relation to learning automata and Q-learning in the context of signaling games, see (Catteeuw & Manderick 2014).

[2] He calls this process of appropriating already-evolved signals as inputs to a game *modular composition*. LaCroix (2020) gives a related model of modular composition in which agents learn to make use of the fixed dispositions of other agents instead of learning them from scratch.

assumed one and only one signal is sent given a state of nature $\sigma$, the two signals together partition the receiver's possible observations. However, in an attention game, the receiver may pay attention to other partitions. Instead of simply observing the signal and then acting, the receiver instead receives a *feature vector*, which is stochastically generated from the signal. A *feature $f$* is a random variable with finite range, and a feature set is a set of features. A feature vector is a vector that specifies the value each feature in the feature set takes in a given instance. We think of the values that a feature can take as partitioning the possible observations of the receiver. The set of possible feature vectors represents the possible observations that the receiver might make.

Instead of the receiver simply conditioning their act $a$ on the value of the signal-feature, the receiver first must choose which value of the feature vector to observe. Nature then determines the payoffs as usual. An attention game differs from a signaling game by introducing on the receiver's side the choice to pay attention to different features, introducing the problem of distinguishing the signal from the noise. In this paper we consider cases in which each feature is binary, so the codomain of each $f_i$ is $\{0, 1\}$. We will always use $f_0$ to denote the signal-feature.

The learning dynamics are constant across all models and are defined as follows. On round $t$, acts in each of the subgames (sender $S$, attention $A$, receiver $R$) conditional on any observations $\sigma_j, s_i$, or $f_k$ are determined probabilistically with Luce's choice rule

$$p^t(x) = \frac{q^t(x)}{\sum_y q^t(y)} \tag{1}$$

where $p^t(x)$ is the probability that the agent takes act $x$ at time $t$. This is a function of the learned weights of all the acts, where $q^t(y)$ is the weight of act $y$ at time $t$.

The update rule for each process is

$$q_A^{t+1}(f_i) = q_A^t(f_i) + \pi_A^t(f_i) \tag{2}$$

$$q_{S,\sigma_j}^{t+1}(s_i) = q_{S,\sigma_j}^t(s_i) + \pi_{S,\sigma_j}^t(s_i) \tag{3}$$

$$q_{R,s_j,f_k}^{t+1}(a_i) = q_{R,s_j,f_k}^t(a_i) + \pi_{R,s_j,f_k}^t(a_i) \tag{4}$$

With initial weights $q_i^0(x) = 1$ for all $x$ and $i$, and payoffs $\pi_i^t$ calculated by (where $\delta$ is the Kronecker delta)

$$\pi_i^t(x) = \begin{cases} \delta_{\sigma^t a^t} & \text{if } x \text{ chosen at } t \text{ with observations specified in } i \\ 0 & \text{otherwise} \end{cases} \tag{5}$$

This learning dynamics, due originally to Richard Herrnstein's work on human and animal learning, can be conceptualized as a simple process of drawing balls from urns. An agent begins with a set of urns corresponding to possible observations. Urns begin with one ball of each color corresponding to possible actions. On making an observation, the agent draws from the corresponding urn and performs the act corresponding to the drawn ball. The agent then returns the ball to the urn and adds a number of balls of the color proportional to the payoff received for performing the action. This algorithm is not identical with the Q-learning algorithm commonly employed in reinforcement learning literature, but is motivated in the philosophical literature by its connection to human learning (e.g. in Erev & Roth (1998)) and to the evolutionary replicator dynamics (Beggs 2005).[3]

## 3    RESULTS

**Uncorrelated Features**    We first consider a $2 \times 2 \times 2$ attention game: each of two states of nature obtains with probability $\frac{1}{2}$, and the sender and receiver each have two possible acts. We suppose that the number of features is at least one, and all other $f_i$ for $i \neq 0$ are determined randomly.[4] We show results for this game under varying parameters in Figure 1. In all versions, the game is set up

---

[3] But see (Barrett & Zollman 2009) and (Catteeuw & Manderick 2014) for comparisons of simple reinforcement learning and Q-learning in the context of signaling games.

[4] The addition of the various features and the attention process means that we cannot apply the Argiento et al. (2009) convergence results to the attention game. Thus, we simulate results which estimate the medium-run performance of learning agents.

so that one feature reflects the signal, and we vary the number of additional uncorrelated features from 0 (a traditional $2 \times 2 \times 2$ Lewis-Skyrms signaling game) up to 4, with $10^3$ simulations of each experimental condition. We find that the cumulative accuracy decreases significantly as the number of non-signal features increases. To use another metric, after $10^6$ plays with 0 non-signal features, 0.996 of the runs ended up with cumulative accuracy above a threshold of 0.75. With 4 non-signal features, this happens on only 0.708 of the runs. Learning is still possible, but the addition of "noise" in the form of uncorrelated features slows down the process.

**Correlated Features** We consider cases in which a single non-signal feature (which we call "the feature" and denote with $f$) now depends on the signal-feature, but not directly on the state of nature. That is, the signal screens off any correlation between the feature and the state of nature. We characterize this dependence with two parameters, $\alpha$ and $\beta$, where $\alpha = P(f = 1 \mid f_0 = 1)$ and $\beta = P(f = 1 \mid f_0 = 0)$. We will use "A" to denote the event $f = 1$, and $s_1$ to denote the event $f_0 = 1$.

The experimental conditions now depend on $\alpha$ and $\beta$. We sample $\alpha$ at intervals of 0.02 on $[0, 1]$, and, for every value of $\alpha$, we sample $\beta$ at intervals of 0.02 on $[0, \alpha]$.[5] We run $10^3$ simulations of each condition, with $10^4$ plays per simulation. Results are given in Figure 2. As the difference between $\alpha$ and $\beta$ decreases, the accuracy of the learned signaling convention decreases correspondingly. On the other hand, the probability that the receiver chooses the correct feature in the attention process increases. This reveals a surprising trade-off. We might think that groups which tend more often to pay attention to the signal-feature would be more successful at learning to perform the correct action. This is the opposite of what we observe.[6]

We use mutual information in order to characterize the extent to which a non-signal feature and a signal-feature are correlated.[7] One subtle issue is that the mutual information depends on the unconditional probabilities of the two random variables. In our context these change as the agents learn up a signaling convention.[8] However, as Figure 3 shows, simulation results show that the mutual information is relatively constant over time. This is in part because $P(s_1 \mid \sigma_1)$ and $P(s_1 \mid \sigma_0)$ sum to 1 both at the beginning of the learning process, and whenever $P(s_1 \mid \sigma_1) + P(s_1 \mid \sigma_0) = 1$, we can calculate the mutual information between the signal-feature and the feature as

$$I(f_0; f) = \frac{\left(\alpha \log \frac{2\alpha}{\alpha+\beta} + \beta \log \frac{2\beta}{\alpha+\beta} + (1-\alpha) \log \frac{2(1-\alpha)}{2-\alpha-\beta} + (1-\beta) \log \frac{2(1-\beta)}{2-\alpha-\beta}\right)}{2} \qquad (6)$$

In Figure 4 we calculate this value for pairs of parameters, $\alpha, \beta \in [0, 1]$. The resulting heat map bears striking similarity to the two heat maps in Figure 2. When we test for the correlation between mutual information and both accuracy and attention, we get almost perfect correlation, specifically $r = 0.991$ between accuracy and mutual information, and 0.988 between attention and mutual information. We see that as the mutual information increases, mistakes in which the receiver pays attention to the less informative feature become less costly. Thus overall accuracy increases. Conversely, as the mutual information increases, this also means that the non-signal feature is reinforced more often in the attention process, which means that the receiver pays more attention to the less informative feature.[9]

**Correlated Features With No Signal-Feature** Finally we investigate attention games in which all features correlate to various degrees with an (unobservable) signal-feature. This attention game

---

[5] No practical difference is made by switching the values of $\alpha$ and $\beta$, so no further conditions are needed.

[6] One might worry about the cumulative accuracy not accurately reflecting the final outcome of learning. We also ran the same simulations with $10^3$ non-learning plays at the end to determine a measure of the final accuracy at the conclusion of the learning process. The results are comparable in all relevant ways.

[7] See Skyrms (2010) for a discussion of the application of information theory to signaling games.

[8] To see this, consider the first term of the sum, $P(A, s_1) \log \frac{P(A, s_1)}{P(A)P(s_1)}$. Notice that this depends on the value $P(A, s_1)$, which we can rewrite as $P(A \mid s_1)(P(s_1 \mid \sigma_1)P(\sigma_1) + P(s_1 \mid \sigma_0)P(\sigma_0)) = \frac{\alpha}{2}(P(s_1 \mid \sigma_1) + P(s_1 \mid \sigma_0))$. The two conditional probabilities, $P(s_1 \mid \sigma_1)$ and $P(s_1 \mid \sigma_0)$, vary as the agents learn a signaling system.

[9] This provides evidence that the agents will learn to signal in the limit, which in turn implies by the results given in (Beggs 2005) that the receiver will attend to the most informative feature in the limit.

would be appropriate for modelling a situation in which the sender has only imperfect control over what the receiver gets to observe. In the case with only one feature this corresponds to a noisy signal-feature. In the case with multiple features, the receiver would have to learn which feature is most informative, and pay attention to it. This is an important case; in the real world, agents will never observe the signal of other agents with perfect fidelity.

In the case in which there is only one feature there is no attention process.[10] We find that success (Figure 5) tracks mutual information ($r = 0.984$). We consider further cases in which the number of features ranges from two to five. To generate the $\alpha$ and $\beta$ parameters for each feature, we use the line segment $\overline{(0.5, 0.5), (1, 0)}$. For the case with $n$ features, we divide this line segment into $n + 1$ sections, and take the coordinates of the dividing points as the values for $\alpha$ and $\beta$. This maximizes the difference of the mutual information between each feature.

In the 2-feature condition, the mean cumulative accuracy was 0.766. For 3 features, 0.790. For 4, 0.797, and for 5, 0.806. The increasing accuracy observed in this experiment is due to the availability of more informative features as our partition of the line segment becomes finer. So, in the 2-feature case, the most informative feature is $\alpha = 0.83, \beta = 0.17$, while in the 5-feature case, it is $\alpha = 0.92, \beta = 0.08$. The relative weight of each feature in the attention process tracks the informativeness of the features (Figure 6), which accounts for the increase in accuracy.

Note that the behavior of the receiver in this model is suboptimal. The optimal behavior is rather to always attend to the most informative feature. In the limit, we conjecture from the results of Beggs (2005) on simple reinforcement learning that the receiver will learn the optimal behavior, as it represents the strongly-dominant pure strategy. What we have shown, then, is that the medium-run behavior of simple learners diverges significantly both from the optimal behavior and the behavior which would be obtained in the limit of learning. It is worth noting as well that this suboptimal behavior resembles probability matching, which is a phenomenon commonly observed in choice experiments involving humans (Vulkan 2000). Icard (2018) shows that, when decision making is costly, there is a connection between Luce's choice rule (eq. 1) and probability matching.

## 4 CONCLUSION

We showed that learning still takes place (albeit more slowly) when there are multiple uncorrelated features and one signal-feature. For correlated features we discovered a surprising trade-off between accuracy and attending to the signal-feature. We showed that the mutual information of the feature and the signal-feature is highly predictive of both accuracy and attention. Finally, we considered cases in which the signal-feature was not one of the observable features. We showed that in the case with only one feature, mutual information once again predicted the success of the agents. In the case with multiple features of varying amounts of information, the receiver learns to pay attention to more informative features.

ACKNOWLEDGMENTS

We thank Jeffrey Barrett for helpful discussions about the models presented here. We also thank Benjamin Genta and Gerard Rothfus for reading a draft of the paper. We also thank the program chairs and two anonymous reviewers for comments.

---

[10] This simple case might also be characterized as a signaling game with a noisy channel. Barrett et al. (2017) consider a similar case in the context of different learning dynamics and a different way of adding noise. Our noise is characterized by the $\alpha$ and $\beta$ parameters, whereas in their model there is a small fixed probability of a random signal.

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

VISUAL APPENDIX

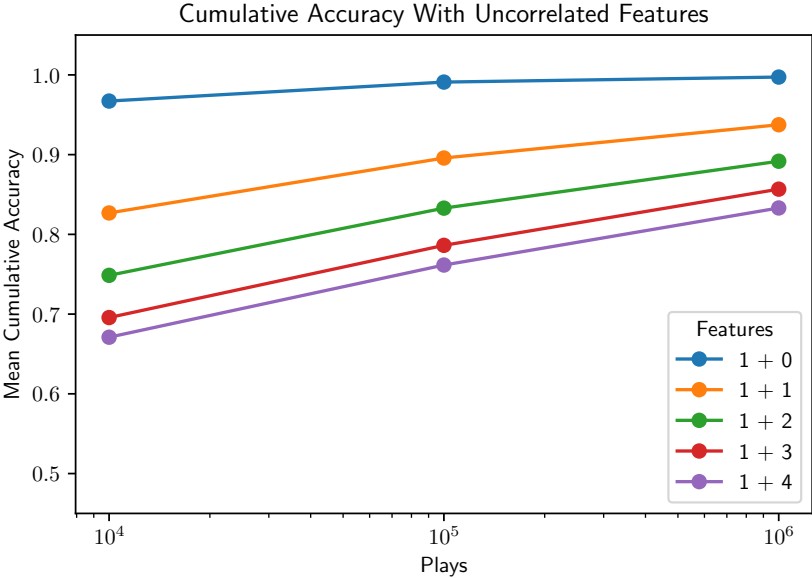

Figure 1: Mean cumulative accuracy for different run lengths (each data point represents the mean of $10^3$ simulations). To emphasize the number of uncorrelated features $i$, we write $1 + i$, with 1 representing the signal feature.

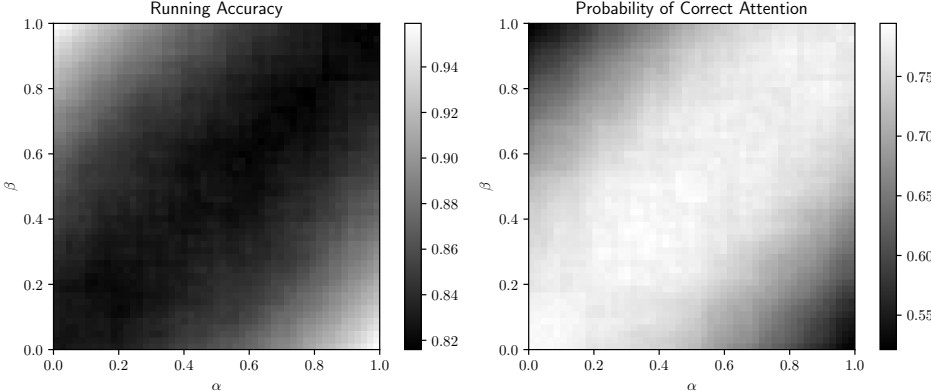

Figure 2: Running accuracy (left) and probability of correct attention (right) for experiments with one correlated feature. Heatmaps are mirrored from upper left to bottom right.

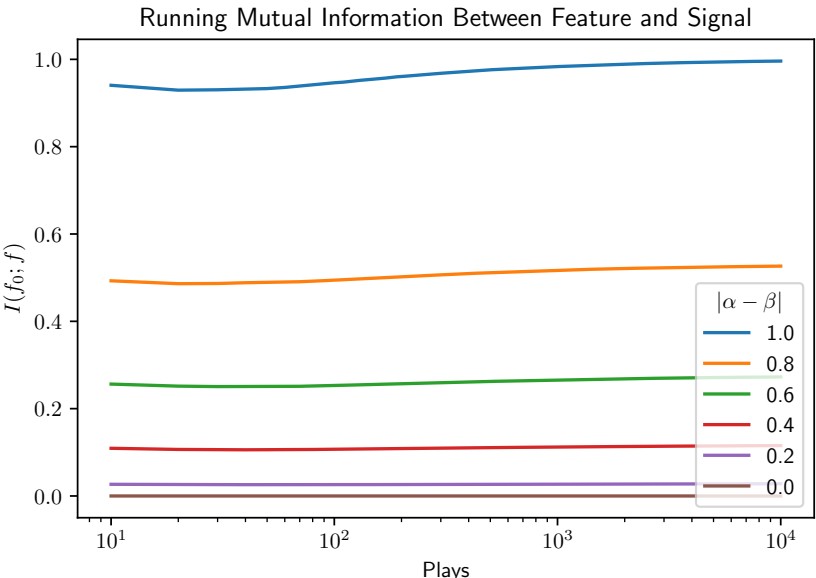

Figure 3: Running mutual information between feature and signal for various degrees of correlation.

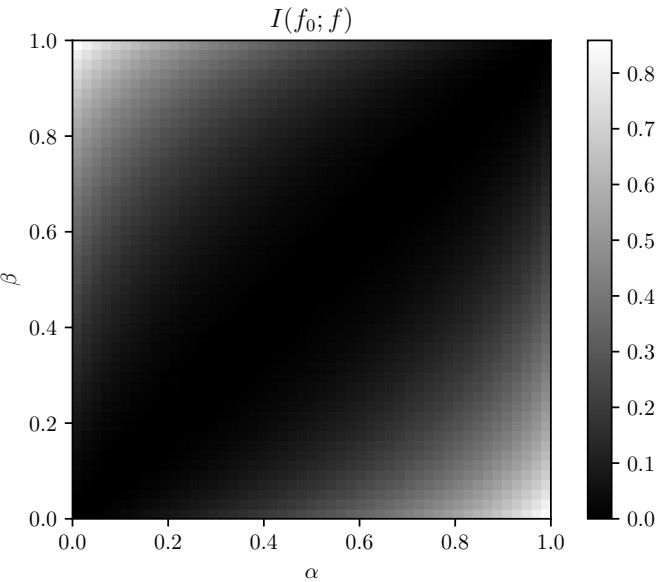

Figure 4: Heatmap of mutual information, measured with intervals of $0.02$ for $\alpha$ and $\beta$.

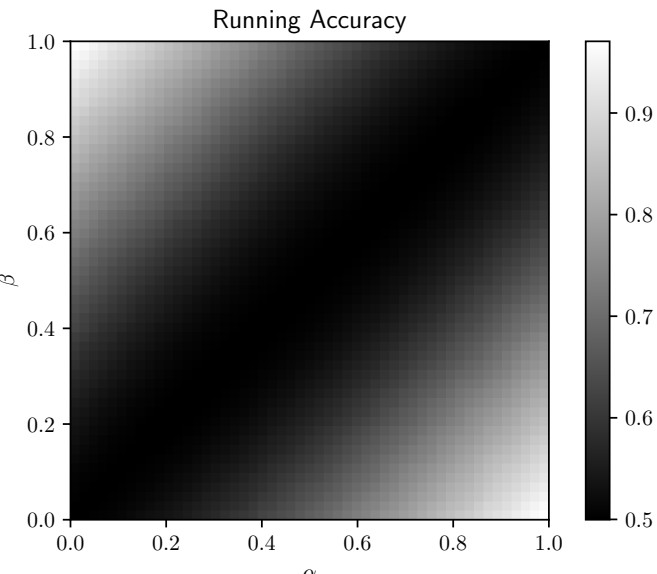

Figure 5: Heatmap of running accuracy (average across $10^3$ simulations run for $10^4$ plays) for experiments with no signal-feature and one correlated feature.

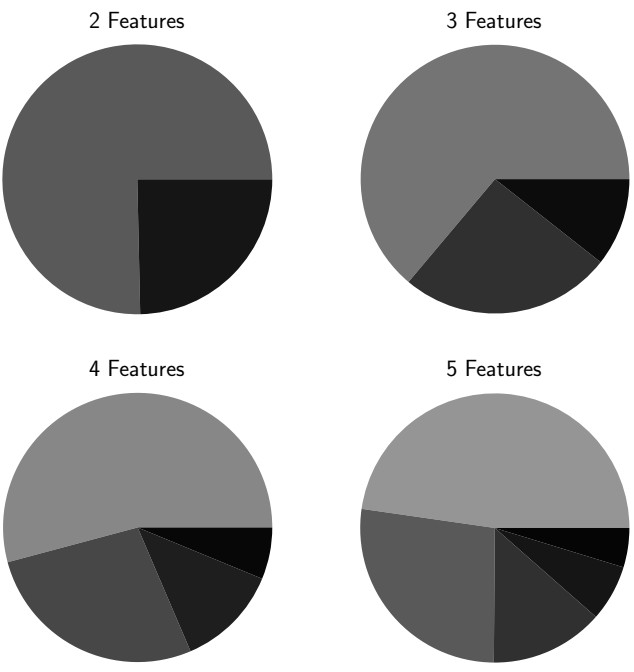

Figure 6: Pie charts showing the proportion of attention given to different features when no signal-feature is available. Lighter shading indicates a higher correlation of the feature (in terms of mutual information) with the unobservable signal-feature.

