# OpenReview forum: "Sifting the Signal from the Noise"
_ICLR.cc/2022/Workshop/EmeCom — EmeCom Workshop at ICLR 2022_

### Official Review · Reviewer_M6Ez · 2022-03-11
**Trivial results with over-complex formulation**

**Rating:** Rejection
**Confidence:** 3

**Review:**

# Summary

## Framework
In this work, the authors investigate a Lewis signaling game where the signal is partitioned in multiple parts, where only a subset is informative.
Differently from Lewis, the receiver receives a feature vector stochastically generated from the signal (called signal-feature or f0).
These two additional elements extend the original game to an attention game where the receiver is also tasked with the choice to pay attention to different features.
The authors use Q-learning to train the sender, attention, and receiver agents.

## Results
This paper presents three different results tied to the presence of noisy features in the signal vector.

### Uncorrelated feature
The authors report a decrease in accuracy when the signal vector contains multiple uncorrelated features (called non-signal features). Precisely, as the number of non-signal features increases, the cumulative accuracy decreases.

### Correlated features
Next, they introduce a non-signal feature (called **the feature** or f) which correlates with the signal-feature but not with the state of nature (i.e., the environment). They focus on two trends: the previously mentioned accuracy and the probability that the receiver chooses the correct feature in the attention process.
Moreover, they define two variables, alpha, and beta, controlling the degree of dependence between f and f0.
They report that, as f and f0 become more independent, the accuracy decreases, and the probability of choosing the correct feature increases.

They also introduce a measure of mutual information between f and f0 and report how the increase in mutual information increases the overall accuracy and the number of times the feature is attended incorrectly.

### Correlated features with no signal-feature
Finally, the authors investigate the setting where no signal-feature is present; instead, some correlated features are used in its place.
They report how an increase in the number of these correlated features also entails increased accuracy.

# Review
Given the previous summary, I find the paper results to be trivial and not valuable for sparking any discussion in the community. In the following sections, I will express my perplexities based on the results paragraphs of the paper.

## Framework
From footnote n1 and expressions [1,2,3,4,5], I guessed the reinforcement algorithm chosen for this task is Q-learning. I would appreciate a more clear formulation of this framework before introducing the above mentioned expressions.

## Uncorrelated feature
It is no surprise how the accuracy of the game will decrease as the number of uncorrelated features increases. Indeed, by adding noisy features to the signal vector, the receiver needs to discern which feature is actually relevant for the game; this task becomes trivially more complicated as the number of noisy feature increases.

## Correlated features
As the author point out, the mutual information between f and f0 positively correlates with the accuracy of the game. I find this straightforward since, when mutual information =1, there is no difference between the amount of information carried by f or f0, thus the choice to pay attention to one or the other does not influence the outcome of the game.
Moreover, the latter also entails a decrease in the probability to choose f0 over f, since they carry the same information about the game.

## Correlated features with no signal-feature
Lastly, while investigating a setting where the singal-feature (f0) is not observable is an interesting direction, the result presented is not remarkable.
When increasing the number of features correlated to f0, more information about f0 is introduced in the signal vector, thus the accuracy of the game increases.

# Conclusion
In my personal opinion, the results present in this paper do not add any relevant point to the emergent communication field. Moreover, I found the paper hard to read since apparently trivial reasoning is presented in an overly complicated formulation.
For the reasons mentioned so far, I chose to reject this work.
I appreciate any kind of answer to this review, especially if the authors find my summary incorrect or imprecise.

---

### Official Review · Reviewer_6MtG · 2022-03-22

**Rating:** Weak accept
**Confidence:** 3

**Review:**

### Summary:
The paper introduces "attention games" as an extension of classic Lewis signaling games that approximate the real-world phenomenon that not all components/features included in a communication protocol are relevant for solving a particular task. To incorporate this into sender-receiver architectures for Lewis signaling games, the paper includes an additional attention module that selects which subset of features from the sender's message should be used by the receiver in decision-making. The paper reports performance of this sender-attention-receiver architecture under three conditions: (i) non-signal features are uncorrelated with the signal feature; (ii) one non-signal feature is correlated with the signal feature; (iii) all features correlate in some way with an unobserved signal feature. Empirical results show that:

 - In case (i), RL agents successfully complete the attention game, but take longer to do so as the number of uncorrelated non-signal features increases.
- in case (ii), RL agents successfully complete the attention game when non-signal features are correlated with the signal feature, so long as mutual information between them is high.
- in case (iii), the success of RL agents when there is no single signal feature (but correlated non-signal features) increases with the number of informative features.

Results are also contextualized through the mutual information between signal and non-signal features in task success, correctly identifying that high accuracy can be attained by attending to non-signal features, so long as the other features have high mutual information with the signal feature.

### Strengths:
- The motivation of the work -- that agents have to sift through features that may or may not be the true signal -- correctly identifies a necessary component of success in emergent communication tasks. In general, emergent signals and signaling games are well-suited to the theme of the workshop.
- The results are well-documented and presented clearly, with ample discussion and formalization.
- The paper situates itself nicely in the context of prior work.

### Questions/Suggestions:
- Equation 1: x, y introduced without definition.
- Section 3, Uncorrelated features: The results clearly document the effect of the number of non-signal features on the speed of learning. Is there anything more concrete/formal that can be said about how the number of plays scales as a function of non-signal features?
- Section 3, Correlated features: "We might think that groups which tend more often to pay attention to the signal-feature..."
    - While it is true that correctly attending to the signal feature will lead to a higher likelihood of success, there might be a bit of a false dichtomy here -- i.e. is this really a surprising trade-off? If mutual information is really high (e.g. close to 1), then attending to the non-signal feature will not impact performance; and conversely for really low mutual information (e.g. close to 0).
    - More generally, is the attention mechanism really attending to features "incorrectly" if it leads to accurate predictions? In this case, high mutual information means that the receiver agent can get the same information by attending to non-signal features as a it does from the signal feature.
    - Is there some case where the mutual information between signal / non-signal features is moderately high. Presenting such an example might help motivate the discussion of this trade-off
- The final paragraph of Section 3 (correlated features with no signal-feature) is the most realistic and interesting scenario -- in most real-world environments, the signal feature is not nicely segmented from other features (as in the previous paragraphs). This also opens the door to study more complex relationships between features. The submission can benefit from more in-depth discussion of these cases.

---

### Decision · Program_Chairs · 2022-03-25

**Decision:**

Accept

**Comment:**

After reading both reviews and reading the paper, we have decided to accept this paper despite the slightly negative reviews. This submission is a work in signalling and some of the negative views may be the disconnect between the fields of ML and signalling. We hope that having the authors at the workshop will allow knowledge sharing sharing between our two fields. We also recommend the authors familiarize themselves with the ML perspectives so that those discussions will be more fruitful (Lazaridou and Baroni's review paper is a good primer).

The paper is a work in signalling evaluating the effect of adding noisy variables and then stochastic noisy signalling. The findings of the paper are quite reasonable and, from the ML perspective, perhaps trivial: adding noise to a channel will make learning slower, attending to highly correlated variables is just as good as attending to the ground truth, and given only correlated variables, the higher the MI then higher the performance. It is notable that these results are well known in ML literature and also in strategic communication literature in economics. If these results are not present in signalling, then it would probably be very useful for the authors to interact with the ML field and also share results from signalling that are not present in ML. For example, I will also clarify that the authors' reinforcement learning algorithm (Erev-Roth or Hernstein) is experimentally linked to human learning has links to Bellman RL's epsilon-greedy Q-learning in tabular environments and policy gradient methods in contextual bandits.

Final note on style: please do not modify the style documents used in construction of the LaTeX. Your submission has both larger spacing, larger margins, and an appendix before the references section. We would like clarify that 4 pages is the maximum for a submission and shorter submissions would be perfectly fine and conforming to style guidelines is more important.